Seed priming enhances seed germination and plant growth in four neglected cultivars of Capsicum annuum L.

Granata Angelo 1
http://orcid.org/0000-0002-9076-7701 Capozzi Fiore 1
Gaglione Anna 1
Riccardi Riccardo 2
Spigno Patrizia 2
Giordano Simonetta 1
Sorrentino Maria Cristina 1 mcristinasorrentino@gmail.com
Spagnuolo Valeria 1 valeria.spagnuolo@unina.it
1 Department of Biology, University of Naples Federico II , Naples , Italy
2 Cooperativa ARCA 2010 , Acerra , Italy
Yasin Nasim
Electronic publication date: 2024 Oct 28
Publication date: 2024
Volume: 12
Electronic Location ID: e18293
Received 2024 Jul 26; Accepted 2024 Sep 21
Copyright: © 2024 Granata et al.
Copyright year: 2024
Copyright holder: Granata et al.
License: This is an open access article distributed under the terms of the Creative Commons Attribution License, which permits unrestricted use, distribution, reproduction and adaptation in any medium and for any purpose provided that it is properly attributed. For attribution, the original author(s), title, publication source (PeerJ) and either DOI or URL of the article must be cited.
License URL: https://creativecommons.org/licenses/by/4.0/

Keywords: Sweet pepper, Germination performance, Hydropriming, Halopriming, Seed vigor index, Development germination index

Funding: Dept. of Biology, Federico II University This work was supported by the Dept. of Biology, Federico II University. The funders had no role in study design, data collection and analysis, decision to publish, or preparation of the manuscript.

==============================
Priming is basically a water-based technique inducing controlled seed rehydration to trigger the metabolic processes normally activated during the early phase of germination. It is regarded as an ecofriendly approach alternative to fertilizers in traditional agriculture, but also a method to synchronize off-field crops and resume stored seeds, improving vigor, and allowing for a rapid, uniform seedling emergence. In this work we tested several methods of seed priming (i.e., hydro-priming, halopriming by KNO3, and acid priming with HCl) in four ancient and neglected cultivars of Capsicum annuum L., a crop species belonging to Solanaceae family cultivated worldwide. We followed germination performance, seedling growth and selected morphological traits, antioxidant production in the leaves, and protein content of the seeds. Apart from acid priming, which inhibited root emergence, both hydropriming and halopriming decreased the mean germination time in all cultivars. The best treatments were KNO3 6% for 96 h > KNO3 4% for 48 h > hydropriming for 24 h. In particular, KNO3 6% for 96 h in all four cultivars significantly increased plant growth, simple vigor index, development germination index, leaf antioxidant concentration and protein content in the seeds, in comparison to control and other priming treatments, indicating the prompt activation of pre-germinative processes.

Introduction

One of the most important challenges in agriculture is reconciling the growing demand for food with several drawbacks, which sometimes derive from the same strategies implemented to increase yield. Among these concerns, poor soil nutrient status, the uncontrolled use of chemicals (fertilizers and pesticides), and the shrinkage of arable soil, are particularly worrying (Paul, Dey & Kundu, 2022).

It is essential to develop alternative practices that ensure high yields while emphasizing sustainability and the responsible management of resources.

In this respect, a good starting point is to improve seed quality by reducing germination time, through seed priming techniques. Priming is basically a water-based technique inducing controlled seed rehydration to trigger the metabolic processes normally activated during the early phase of germination (the so-called pre-germinative metabolism). Priming improves product quality directly affecting seed vigor, a complex agronomic trait controlled by multiple genetic and environmental factors (Jisha, Vijayakumari & Puthur, 2013; Paparella et al., 2015; Rajjou et al., 2012). Achieving rapid and uniform seedling emergence is, indeed, a key point for crop yield. Moreover, slow germination rates frequently expose vulnerable plantlets to adverse environmental conditions and soil-borne infections (Paparella et al., 2015). Seeds subjected to priming show increased germination rates, which result in high levels of resistance to biotic/abiotic stress and the improvement of crop yield (Ajouri, Asgedom & Becker, 2004; Ibrahim, 2016; Marthandan et al., 2020). To limit the use of chemicals in agricultural practices, pre-sowing techniques like seed priming, attract the attention of researchers since these techniques simultaneously addresses the problems of slow and non-uniform seed germination, low seed vigor, poor harvest and product quality, (Chatterjee et al., 2018; Devika et al., 2021; Paparella et al., 2015; Sarkar & Rakshit, 2020; Zulfiqar, 2021). At the microscopic level, beneficial effects of this technology are the preservation of cell membrane integrity by protection of lipids from peroxidation, the antipathogenic effects, the repair of biochemical damages by the enzymatic repair system and the metabolic degradation of toxic compounds (Mondal & Bose, 2014).

Seed priming techniques mirror natural priming strategies. In nature indeed, seeds that undergo natural burial in the soil, are exposed to a variety of edaphic factors like changes in pH, soil texture and humidity, and daily fluctuations of temperature, light, and oxygen, which positively regulate germination (González-Zertuche et al., 2001; Paul, Dey & Kundu, 2022). Thus, the burial of seeds along with soil microenvironments, impose a natural priming effect speeding up seed physiology and shoot emergence. Moreover, seed burial is also beneficial for those places experiencing regular fires that could eventually kill plant organisms, but preserving the local seed bank (Wijayratne & Pyke, 2012). Further, priming can be used as a biologically important technique in the domestication of medicinal plants About half of pharmaceuticals (e.g., Codeine, Digoxin) used nowadays are derived by studies focused on the effects of medicinal plants, most of which have difficulty in seed germination and seedling establishment in the field (Chakraborty & Dwivedi, 2021). Priming can be also a valuable tool to resume the culture of a given species for ex situ conservation of germplasm collections (Paparella et al., 2015). In this respect, the recent Report FAO on Global State of Biodiversity for Food and Agriculture (FAO, 2019), underlines the role that food production systems play in the loss of biodiversity, including agricultural biodiversity. This document highlights the sudden decline and disappearance on a global scale of numerous local varieties of cultivated plants and wild parental species, many of which are maintained by farmers mainly for self-consumption. Specifically, Italy boasts a conspicuous heritage of traditional horticultural species and varieties. This heritage also includes varieties widely used in the past, and now limited to a few private gardens; others that are well known because they are protagonists in the commercial and culinary fields (e.g., for tomato, the “piennolo” from Vesuvius, or “pachino” from Sicily, “costoluto” from Tuscany, along with many others). The conservation of these varieties involves profoundly different geographical areas, representing a cultural, economic, nutritional, botanical, and ecological heritage. Therefore, the protection of horticultural biodiversity is of fundamental importance, but it is equally important to achieve this objective while respecting the agricultural ecosystem, which must be both productive and sustainable.

Based on the above, aim of the present work is to improve germination performance in four ancient and neglected cultivars of Capsicum annuum L. cultured in Campania region (southern Italy), showing a rapid decrease in germination percentage during aging. Under the hypothesis that specific priming procedures could improve seed germination performance and plant growth in these cultivars, we tested several methods of seed priming (i.e., hydro-priming, halo-priming, and acid priming) and followed (i) germination performance; (ii) seedling growth and morphological traits; (iii) antioxidant production; (iv) protein content.

Materials and Methods

Plant material

Capsicum annuum L. (Solanaceae) is an annual plant native of the South America, arrived in Europe, together with tomato, in the 16th century. It has an elongated floral axis with 1 flower per node. The flowers have a white perianth and the fruit (with a variable color from ivory to purple, but in most varieties from yellow to red) is a berry about fifteen centimeters long, with numerous discoidal seeds with a diameter of 3–5 mm. The fruits of C. annum are known for their antioxidant properties, thanks to the presence of vitamin C and beta-carotene, which help protect cells from oxidative damage, furthermore, they are rich in vitamin A, potassium, and fiber.

C. annuum is widespread in Campania, and its fruits are considered one of the main vegetables grown in the region. Campania is famous for its traditional cultivars of peppers, including “sassaniello” and “papaccella”. For the present study we examined four ancient and neglected cultivars (i.e., cultivated by few households, or on small areas; Orobiyi et al., 2017): sassaniello giallo (SY), sassaniello rosso (SR) (i.e., yellow and red sassaniello), papaccella liscia gialla (PY), and papaccella liscia rossa (PR) (i.e., smooth yellow and red papaccella). Seeds were provided by the cooperative company Arca 2010, a germplasm bank of Campania region.

Priming treatments

Seeds provided by Arca 2010 were obtained from the ripe fruit, dried in the sun, and stored 1 month at +4 °C until they were used. The experiments were carried out in the period October 2021–January 2022. One gram seeds (about 120 seeds) for each cultivar were subjected to each priming treatments: halopriming with KNO3 at three different concentrations (2%, 4%, 6%) and for three different times (24, 48, 96 h); hydropriming for 24, 48, 96 h; acid priming (100 mM HCl) for 5, 10 and 15 min, for a total of 15 priming treatments; further 0.5 g of untreated seeds were tested as control. We chose these acid priming and halopriming with KNO3 according to literature reports (Agoncillo, 2018; Kumar, Chaurasia & Marmat Sandip, 2021; Quintero et al., 2018; Venkatasubramanian & Umarani, 2010). For each treatment seeds were submerged in water or aqueous solution for the selected times. Then, seeds were dried in oven at +30 °C, until constant weight.

Seed vitality assessment

Seed vitality was evaluated by 2,3,5-triphenyl-tetrazolium chloride (TTC) test after priming and re-drying of the seeds. The tetrazolium test indirectly verifies the respiratory activity in the cells of the seed tissues. The test is based on the activity of dehydrogenase enzymes, and particularly malate dehydrogenase; this latter by oxidation of its substrate, determines the release of electrons and hydrogen ions, which reduce the TTC in living tissues. The reduced form of the TTC-salt is a red-colored, stable, non-diffusible substance called triphenyl-formazan (França-Neto & Krzyzanowski, 2022). Specifically, 20 seeds for each cultivar and treatment were put in Petri dishes on wet paper added of 5 mL distilled water at +20 °C for 18 h. Seeds were then cut along the embryonic axis and immersed in 1% tetrazolium chloride solution (in phosphate buffer) for 6 h at +30 °C, in the dark (Kusumawardana, Pujiasmanto & Pardono, 2018). Seeds were observed under a stereomicroscope at 40x magnification.

Germination and growth

For each germination test 30 seeds were used, and the experiments were carried out in triplicate. After the priming treatment and drying, the seeds were germinated in Petri dishes on wet filter paper in the dark at +20 ± 1 °C, until the seedling development. During this period the following parameters were calculated: the germination percentages (GP) at 5, 10 and 15 days (i.e., the percentage of germinated seeds with 2 mm emerged radicle); the mean germination time (MGT); T50 index, i.e., the time (days) necessary for 50% of viable seeds to germinate; the simple vigor index (SVI); the germination development index (DGI).

The mean germination time (MGT) was calculated using the following formula (Ellis & Roberts, 1979):

MGT=∑Dn∑n

where n is the number of seeds germinated on day D and D is the number of days counted from the start of germination at each count. The denominator is the summation of the seeds germinated at each count; therefore, it is equal to the total number of germinated seeds.

The parameter T50 is the time needed for 50% of the final germination percentage (FGP), and together with MGT allows to estimate the germination speed.

The simple vigor index (SVI) (Tu et al., 2022; Wang et al., 2021).

SVI=FW×FGP.

Equal to the product of the total fresh weight of the seedlings (FW) for the final percentage of germination (FGP).

The development germination index (DGI) was calculated following the formula (Quintero et al., 2018):

100×(FGP(t)FGP(0))×(PRL(t)PRL(0)).

Equal to the percentage product between FGP observed in primed seeds compared to control seeds, and primary radicle length observed in primed seeds compared to control seeds.

Where FGP (t) is the final germination percentage of primed seeds, FGP (0) is the final germination percentage observed in control seeds, PRL (t) is the primary root length of treated seeds, and PRL (0) is the length of the primary root of control seeds. Of course, being expressed in comparison to control, DGI is equal to 1 (or 100%) in the control; DGI > 1 (100%) indicates that priming treatment improve germination performance, and vice versa.

The seedlings were grown in seedbeds on vermiculite, at the temperature of 25/20 °C (day/night), relative humidity (RH) of 60–75% (day/night) and a photoperiod of 16 h of light per day with a photosynthetic photon flux density (PPFD) at the top of the canopy of 180–190 μmol photons m−2 s−1. Seedlings were watered with Murashige Skoog salt liquid medium diluted with water 1:1. At 40 days of culture, the following parameters were measured: total fresh and dry weight of the plants, the weight of the shoots and roots, the length of the roots and the height of the plants.

Morphological observations

Morphological traits considered in previous studies (e.g., Kumar, Chaurasia & Marmat Sandip, 2021; Venkatasubramanian & Umarani, 2010), such as root length and fresh weight, shoot length and fresh weight, total fresh weight, total dry weight, and leaf length (second leaf stage), were measured in 10 plants per species and treatment at 40 days culture. The shoot height was measured from the collar to the end of the highest leaf; the root length was measured from the collar to the longest tip of the primary root. The dry weight was measured after oven treatment at +60 °C until constant weight.

Total antioxidants

Total antioxidants were estimated in the leaves of the four cultivars showing the best germination and growth performance (i.e., Hydropriming for 24 h, KNO3 4% for 48 h, and KNO3 6% for 96 h), and in relative controls, by the ferric reducing antioxidant power (FRAP) assay (Sorrentino et al., 2023). In brief, fresh leaf samples (250 mg), were homogenized and resuspended with a 60:40 (v/v) methanol/water solution; then, the suspension was centrifuged at 14,000 rpm for 15 min at 4 °C; once recovered, the extract was added to 300 mM acetate buffer containing 10 mM tripyridyltriazine (TPTZ) in 40 mM HCl (1:1.6), and 12 mM FeCl3 (1:16), with a final concentration 1:20 (extract: buffer) and pH 3.6. After 1 h incubation at 4 °C, the absorbance at 593 nm was measured with a spectrophotometer (UV-VIS Cary 100; Agilent Technologies, Palo Alto, CA, USA) using Trolox (6-hydroxy-2,5,7,8-tetramethylchroman-2-carboxylic acid) as standard. The total antioxidant capacity was expressed as μmol Trolox equivalents for mg of fresh tissue.

Protein content

Protein content was estimated in the primed seeds of the four cultivars showing the best germination and growth performance (i.e., Hydropriming for 24 h, KNO3 4% for 48 h, and KNO3 6% for 96 h), in addition to untreated seeds as control. For protein extraction seeds (300 mg) were pulverized and suspended in a solution containing 10% tri-chloro-acetic acid (TCA) and 80% methanol in 100 mM ammonium acetate. The suspension was centrifuged at 13,000 rpm at +4 °C for 3 min at each addition; finally, the pellet was recovered and dried in oven at +50 °C for 10 min. Then phenol and SDS buffer were added for pellet resuspension. After this step, samples were centrifuged for 10 min at 13,000 rpm and room temperature, to separate the phenol phase containing protein fraction. After adding 80% methanol in ammonium acetate samples were stored overnight at −20 °C. Afterwords, samples were centrifuged at 13,000 rpm for 10 min and the pellet obtained was washed with 100% methanol and 80% acetone two times. Finally, the protein pellet was suspended in a buffer containing a protease inhibitor cocktail (Sigma) and phenyl-methyl-sulfonyl fluoride (PMSF) in 0.1% ethanol. The protein content was determined according to the Bradford method (1976), based on the binding between protein and Coomassie Brilliant Blue G-250, measuring absorption at 595 nm, which is proportional to the protein content.

Data analysis

The data were processed using Microsoft Excel and IBM SPSS for Windows (IBM Corp. Released 2020, Version 27.0. Armonk, NY, USA). As for the percentage data, the analysis was performed after arcsine transformation. Generalized linear models (GLMs) were applied to MGT, SVI, DGI, T50 and morphological data (considered as dependent variables) to test the main and the interactive effects of cultivar (fixed effect with four levels: PR, PY, SR and SY), treatments (fixed effect with five levels: halopriming with three different concentrations −2%, 4%, 6%, hydropriming and control), time (fixed effect with three levels: 24, 48, 96 h). For the priming treatments showing the best germination and growth performance, the differences between the means of total antioxidant and protein content were tested by ANOVA. As post hoc test, the Tukey’s HSD was used with a significant level for p < 0.05.

Results

Seed vitality and germination

The observation of control and primed seeds after tetrazolium test showed red-colored seed tissues in all control and treated seeds apart from those seeds subjected to acid priming. Here, regardless the treatment times, tissues appeared white or poorly colored both in the embryo and in the endosperm (Fig. S1). Due to this result, acid priming was excluded from further analyses.

The result of the GLM (Table S1) applied to MGT showed a significant interaction between cultivars, treatments and times. All priming treatments significantly decreased the MGT compared to control, with a reduction overall cultivar higher under KNO3 4% and 6% for 48 and 96 h, respectively.

The MGT under the different treatments and times grouped for each CV is illustrated in Fig. 1. In control seeds the MGT was higher in papaccella compared to sassaniello. Specifically, in PR the highest decrease of MGT was observed with KNO3 4% for 48 h and KNO3 6% for 96 h; in PY the highest reduction of MGT was recorded in seeds primed by KNO3 4% for 48 h; in SR the highest reduction of MGT occurred under KNO3 4% for 48 h and KNO3 6% for 96 h treatments, whereas in SG only by KNO3 4% for 48 h.

Figure 1 Average mean germination time (MGT).

Bar charts of the average mean germination time (MGT) of papaccella ((A) PR; (B), PY) and sassaniello ((C), SR; (D), SY) for control (C), hydropriming (H) and halopriming (KNO3, 2%, 4% and 6%) at 24, 48 and 96 h. The error bars are standard deviations. Different letters indicate significant differences according to Tukey’s HSD test (p < 0.05).

Morphological observation

The results of the GLMs (Table S1) applied to morphological data highlighted a significant interaction between cultivars, treatments and times. Morphological traits measured at 40 days culture (Table 1) showed that all plant characters were affected by seed priming treatments. Specifically, in SR the most promising treatments were KNO3 2% for 48–96 h and KNO3 6% for 96 h; the first significantly increased root length and weight and total weights; the latter significantly increased all measured traits. In SY instead, hydropriming for 24 h, and KNO3 6% for 48 and 96 h significantly increased all selected traits. In PR and PY most traits significantly increased in plants grown from seeds subjected to KNO3 4% for 48 h and KNO3 6% for 48 and 96 h.

Table 1 Morphological traits at 40 days after transplant (mean ± standard deviation).

		Root weight (g)		Shoot weight (g)		Total weight (g)		Root length (cm)		Shoot height (cm)		Leaf length (mm)				Root weight (g)		Shoot weight (g)		Total weight (g)		Root length (cm)		Shoot height (cm)		Leaf length (mm)		
SR	H_24	0.5 ± 0.3	a	0.8 ± 0.4	ab*	1.3 ± 0.7	ab	17.1 ± 3.8	a	4.4 ± 0.9	ab	37.4 ± 5.8	b	SY	H_24	0.2 ± 0.1	b	0.4 ± 0.2	bc	0.7 ± 0.2	bc	9.8 ± 2.5	c	3.6 ± 0.4	ab	34.1 ± 5.8	ab	
H_48	0.3 ± 0.1	a	0.60 ± 0.3	ab	1.0 ± 0.4	ab	13.2 ± 4.8	ab	3.7 ± 0.4	b	41 ± 11	ab	H_48	0.3 ± 0.2	ab	0.5 ± 0.2	bc	0.7 ± 0.4	abc	15.1 ± 1.9	b	4.8 ± 0.8	a	45.5 ± 9.8	a	
H_96	0.2 ± 0.1	ab	0.3 ± 0.1	c	0.4 ± 0.1	d	14.2 ± 2.4	ab	3.3 ± 1.2	ab	38 ± 6.5	b	H_96	0.1 ± 0.0	c	0.3 ± 0.1	c	0.4 ± 0.2	c	11.9 ± 6.8	bc	2.7 ± 1.1	b	43 ± 9.9	a	
K_2%_24	0.4 ± 0.2	a	0.5 ± 0.2	bc	0.9 ± 0.3	bc	17.5 ± 1.9	a	4.5 ± 1.2	ab	41.5 ± 9.6	a	K_2%_24	0.4 ± 0.1	ab	0.8 ± 0.2	a	1.2 ± 0.3	a	20.2 ± 1.7	a	5.4 ± 0.6	a	37.5 ± 6.7	ab	
K_2%_48	0.4 ± 0.1	a	0.8 ± 0.1	ab	1.2 ± 0.2	a	18.4 ± 3.4	a	4.7 ± 0.9	ab	45 ± 5.3	a	K_2%_48	0.3 ± 0.0	b	0.5 ± 0.1	b	0.8 ± 0.1	b	16.0 ± 2.6	ab	3.9 ± 0.6	ab	35.3 ± 7.6	ab	
K_2%_96	0.4 ± 0.2	a	0.8 ± 0.3	ab	1.1 ± 0.4	a	16.8 ± 0.9	a	4.4 ± 0.6	ab	44.7 ± 11	a	K_2%_96	0.3 ± 0.1	ab	0.5 ± 0.1	b	0.8 ± 0.1	b	14.6 ± 2.1	b	3.8 ± 0.7	ab	34.2 ± 7.9	b	
K_4%_24	0.2 ± 0.1	ab	0.4 ± 0.1	bc	0.5 ± 0.2	cd	15.2 ± 4.6	ab	4.6 ± 1.1	ab	39 ± 7.3	ab	K_4%_24	0.3 ± 0.2	ab	0.5 ± 0.3	abc	0.9 ± 0.5	abc	11.1 ± 3.2	bc	3.7 ± 0.8	ab	37.4 ± 6.8	ab	
K_4%_48	0.2 ± 0.1	ab	0.3 ± 0.2	c	0.5 ± 0.3	cd	14.3 ± 2.1	ab	4.5 ± 0.8	ab	43.6 ± 6.5	a	K_4%_48	0.5 ± 0.1	a	0.8 ± 0.1	ab	1.3 ± 0.1	a	15.5 ± 3.1	ab	5.1 ± 0.7	a	41.6 ± 10	ab	
K_4%_96	0.3 ± 0.2	ab	0.5 ± 0.2	bc	0.8 ± 0.5	bc	14.3 ± 2.8	ab	4.3 ± 0.8	ab	35.6 ± 5	ab	K_4%_96	0.4 ± 0.1	ab	0.8 ± 0.2	ab	1.3 ± 0.3	a	14.8 ± 1.9	b	4.7 ± 0.5	a	36.2 ± 3.4	ab	
K_6%_24	0.3 ± 0.2	ab	0.5 ± 0.2	bc	0.7 ± 0.3	ab	12.8 ± 4.9	ab	4.4 ± 1.3	ab	42 ± 9.1	a	K_6%_24	0.3 ± 0.2	ab	0.6 ± 0.4	b	0.9 ± 0.6	abc	11.6 ± 3.3	bc	3.8 ± 0.8	ab	39.4 ± 6.7	ab	
K_6%_48	0.4 ± 0.1	a	0.7 ± 0.3	ab	1.1 ± 0.4	ab	16.3 ± 2.6	a	4.6 ± 1.4	ab	52.8 ± 8	a	K_6%_48	0.4 ± 0.1	ab	0.9 ± 0.1	a	1.4 ± 0.2	a	14.8 ± 1.4	b	4.8 ± 0.6	a	49.1 ± 4.7	a	
K_6%_96	0.4 ± 0.1	a	1.1 ± 0.3	a	1.4 ± 0.4	a	17.5 ± 1.5	a	5.3 ± 0.7	a	57.8 ± 4.4	a	K_6%_96	0.6 ± 0.2	a	0.9 ± 0.3	a	1.5 ± 0.4	a	18.7 ± 4.1	ab	5.6 ± 1.3	a	56.2 ± 6.9	a	
	C	0.1 ± 0.0	b	0.3 ± 0.1	c	0.4 ± 0.1	d	10.5 ± 3.4	b	2.8 ± 0.9	b	36.5 ± 3	b		C	0.1 ± 0.0	c	0.3 ± 0.1	c	0.4 ± 0.2	c	9.8 ± 3.2	c	2.8 ± 0.8	b	33.3 ± 2.5	b	
PR	H_24	0.5 ± 0.1	a	0.9 ± 0.2	a	1.4 ± 0.4	a	20.2 ± 1.6	a	5.8 ± 0.9	a	30.5 ± 2.7	b	PY	H_24	0.2 ± 0.1	bc	0.3 ± 0.1	c	0.5 ± 0.2	b	11.6 ± 3.7	b	3.4 ± 0.4	b	30.1 ± 4.3	b	
H_48	0.4 ± 0.1	a	0.7 ± 0.3	ab	1.2 ± 0.5	a	13.6 ± 4.4	b	4.1 ± 0.6	b	37.7 ± 1.2	ab	H_48	0.4 ± 0.2	ab	0.7 ± 0.5	ab	1.1 ± 0.7	ab	21.5 ± 4.4	ab	4.3 ± 1.4	b	38 ± 8.4	a	
H_96	0.3 ± 0.1	a	0.7 ± 0.2	ab	1.1 ± 0.3	a	12.8 ± 3.7	b	4.3 ± 0.7	ab	34.4 ± 6.6	b	H_96	0.3 ± 0.1	ab	0.5 ± 0.3	bc	0.8 ± 0.4	ab	14.2 ± 6.1	b	4.0 ± 0.7	b	38 ± 7.4	a	
K_2%_24	0.3 ± 0.1	a	0.6 ± 0.3	ab	0.9 ± 0.5	ab	17.6 ± 2.9	ab	4.2 ± 0.9	ab	39.8 ± 4.8	b	K_2%_24	0.3 ± 0.2	abc	0.7 ± 0.3	ab	1.1 ± 0.5	a	19.1 ± 5.4	ab	4.9 ± 0.8	ab	35.2 ± 7.3	ab	
K_2%_48	0.3 ± 0.2	ab	0.5 ± 0.2	b	0.8 ± 0.3	b	16.3 ± 1.7	ab	3.6 ± 1.2	b	37.9 ± 5.9	ab	K_2%_48	0.2 ± 0.1	bc	0.5 ± 0.1	bc	0.7 ± 0.2	ab	13.0 ± 3.4	b	3.7 ± 0.9	b	34.9 ± 4.5	ab	
K_2%_96	0.3 ± 0.1	ab	0.4 ± 0.2	b	0.7 ± 0.4	b	11.6 ± 4.6	b	3.4 ± 0.5	b	33.1 ± 3.4	b	K_2%_96	0.3 ± 0.2	abc	0.6 ± 0.4	abc	0.9 ± 0.6	ab	16.4 ± 2.8	b	3.5 ± 0.7	b	37.6 ± 4.3	ab	
K_4%_24	0.3 ± 0.1	ab	0.6 ± 0.2	b	0.9 ± 0.3	a	14.0 ± 4.2	b	3.6 ± 0.4	b	32.7 ± 1.3	b	K_4%_24	0.4 ± 0.1	ab	0.6 ± 0.1	abc	1.1 ± 0.2	a	16.1 ± 5.5	b	4.5 ± 0.6	b	35.1 ± 6.6	ab	
K_4%_48	0.4 ± 0.1	a	0.8 ± 0.3	ab	1.1 ± 0.5	a	16.2 ± 3.4	ab	4.5 ± 0.7	ab	39 ± 4.4	ab	K_4%_48	0.4 ± 0.2	ab	0.8 ± 0.3	ab	1.3 ± 0.4	a	21.0 ± 3.7	ab	5.4 ± 1.2	ab	34.7 ± 2	ab	
K_4%_96	0.2 ± 0.1	b	0.4 ± 0.1	b	0.6 ± 0.2	b	14.8 ± 4.4	b	3.3 ± 0.5	b	29.9 ± 5.3	b	K_4%_96	0.2 ± 0.1	bc	0.4 ± 0.2	bc	0.6 ± 0.4	b	9.2 ± 2.5	b	3.0 ± 0.8	b	29.9 ± 5.3	b	
K_6%_24	0.4 ± 0.1	a	0.7 ± 0.2	ab	1.1 ± 0.3	a	16.7 ± 1.6	ab	4.1 ± 0.8	b	29.2 ± 2.3	b	K_6%_24	0.4 ± 0.1	ab	0.7 ± 0.2	ab	1.2 ± 0.3	a	14.8 ± 2.1	b	3.6 ± 0.9	b	27.2 ± 3.7	b	
K_6%_48	0.5 ± 0.3	ab	0.9 ± 0.3	a	1.5 ± 0.6	a	16.5 ± 1.8	ab	4.8 ± 0.9	ab	44.4 ± 9.5	ab	K_6%_48	0.4 ± 0.1	ab	\1	ab	1.2 ± 0.3	a	16.1 ± 4.8	b	4.2 ± 0.7	b	38.8 ± 4.1	a	
K_6%_96	0.5 ± 0.2	ab	0.9 ± 0.2	a	1.5 ± 0.3	a	21.1 ± 4.9	ab	6.1 ± 1.5	ab	53.5 ± 9.5	a	K_6%_96	0.6 ± 0.2	a	0.9 ± 0.3	ab	1.5 ± 0.4	a	23.2 ± 2.9	a	6.1 ± 0.7	a	38.4 ± 2.4	a	
	C	0.2 ± 0.1	b	0.4 ± 0.1	b	0.5 ± 0.1	b	13.5 ± 3.7	b	3.3 ± 0.8	b	30.6 ± 1.8	b		C	0.1 ± 0.0	c	0.3 ± 0.1	c	0.4 ± 0.1	b	9.9 ± 4.3	b	3.0 ± 0.7	b	30.7 ± 1.7	b	
Note:

* Different letters indicate significant differences according to Tukey’s test, p < 0.05.

In general, all seed priming boosted plant development, inducing a significant increase in biomass production (i.e., total weight). The different contribution of root and shoot weights to total weight is reported as well (Table 1). In all cultivars priming treatments affected both hypogeal and epigeal weight, with a more pronounced effect in plants from seeds primed by KNO3, compared to hydropriming. Specifically, by comparing the total weight measured in control plants and in plants from seeds primed with KNO3 6% for 96 h, a significant increase was found in the latter in all four cultivars, highlighting that this treatment seems overall the most reliable.

Indexes related to germination performance

The results of the GLMs (Table S1) applied to the indexes related to germination indicated a significant interaction between the cultivars, treatments and times. Indexes related to germination performance (DGI, T50 and SVI) are reported in Table 2 for all cultivars and treatments. In SR DGI and SVI provided the best values under hydropriming 24 h, KNO3 2% for 24–48 h and KNO3 6% for 96 h priming treatments; T50 instead, was about halved by KNO3 4% for 24–48 h and KNO3 6% for 96 h. A similar trend was observed in SY. In both PR and PY the priming by KNO3 6% for 96 h provided the best performance, since it significantly increased DGI and SVI and strongly reduced T50.

Table 2 Indexes of germination performance.

		DGI (%)		T50 (d)		SVI				DGI (%)		T50 (d)		SVI		
SR	H_24	169.7 ± 17.1	a*	12 ± 0	ab	115.9 ± 32	ab	SY	H_24	174.2 ± 29.9	a	10.0 ± 0.1	b	155.2 ± 17.2	a	
H_48	132.8 ± 11.2	bc	11 ± 0	ab	108.0 ± 21.4	ab	H_48	113.6 ± 11.2	b	10.0 ± 0.2	b	117.1 ± 28.2	ab	
H_96	111.3 ± 11.7	c	14 ± 0	a	36.7 ± 13.8	c	H_96	86.07 ± 22.1	c	12.1 ± 0.1	a	90.46 ± 22.5	b	
K_2%_24	164.6 ± 17.8	ab	10 ± 0	b	83.3 ± 26.2	bc	K_2%_24	133.5 ± 19.9	b	11.1 ± 0.0	a	84.54 ± 23.7	b	
K_2%_48	168.6 ± 22.9	ab	9 ± 0	b	125.8 ± 20.2	ab	K_2%_48	122.6 ± 15.3	b	9.0 ± 0.1	b	92.2 ± 16.7	b	
K_2%_96	121.2 ± 9.96	c	10 ± 0	b	104.2 ± 23.5	ab	K_2%_96	104.2 ± 16.8	bc	9.1 ± 0.3	b	88.71 ± 12.4	b	
K_4%_24	138.1 ± 19.6	bc	7 ± 0	c	52.6 ± 18.7	c	K_4%_24	119.2 ± 13.8	b	9.1 ± 0.1	b	97.26 ± 22.5	b	
K_4%_48	134.7 ± 19.9	bc	7 ± 0	c	55.1 ± 29.9	c	K_4%_48	123.7 ± 13.5	b	5.2 ± 0.2	c	155.4 ± 25.4	a	
K_4%_96	134.9 ± 9.73	bc	11 ± 0	ab	69.4 ± 33.1	bc	K_4%_96	117.2 ± 11.5	b	9.0 ± 0.1	b	61.64 ± 12.4	b	
K_6%_24	141.7 ± 10.9	abc	11 ± 0	ab	78.9 ± 23	bc	K_6%_24	129.5 ± 15.8	b	10.8 ± 0.7	ab	112.8 ± 26.4	ab	
K_6%_48	155.2 ± 18.6	ab	9 ± 0	b	124.3 ± 28.4	ab	K_6%_48	132.8 ± 29.3	ab	9.0 ± 0.0	b	132.9 ± 11.8	a	
K_6%_96	164.6 ± 14.1	ab	6 ± 0	c	147.5 ± 19.1	a	K_6%_96	171.8 ± 33.9	a	10.0 ± 0.0	b	150.4 ± 20.3	a	
	C	100.0 ± 0.0	c	11 ± 0	ab	49.6 ± 18.2	c		C	100.0 ± 0.0	c	12.0 ± 0.0	a	50.7 ± 14.3	c	
PR	H_24	87.87 ± 13.8	c	11.0 ± 0.0	b	58.79 ± 12.4	c	PY	H_24	101.1 ± 21	c	11.0 ± 0.0	ab	47.37 ± 14.5	cd	
H_48	118.1 ± 24.7	bc	13.0 ± 0.0	b	74.37 ± 25.9	bc	H_48	209.7 ± 29.1	a	12.0 ± 0.0	a	91.79 ± 19.6	bc	
H_96	120.9 ± 14.0	b	14.0 ± 0.0	b	39.05 ± 10.3	c	H_96	146.3 ± 44.5	abc	13.0 ± 0.0	a	67.92 ± 19.6	c	
K_2%_24	165.8 ± 32.7	ab	11.0 ± 0.0	b	115.9 ± 15.9	ab	K_2%_24	148.6 ± 31.8	b	10.0 ± 0.0	b	94.6 ± 26.1	bc	
K_2%_48	136.6 ± 24.3	b	8.0 ± 0.0	c	72.79 ± 14.9	bc	K_2%_48	123.5 ± 24.2	bc	9.0 ± 0.0	b	69.65 ± 19.6	c	
K_2%_96	110.7 ± 22.3	bc	11.0 ± 0.0	b	71.76 ± 11.5	bc	K_2%_96	158.5 ± 31.9	ab	10.0 ± 0.0	b	65.82 ± 18.9	c	
K_4%_24	87.26 ± 21.2	c	11.0 ± 0.0	b	111.2 ± 21.3	ab	K_4%_24	137.0 ± 31.4	b	10.0 ± 0.0	b	103.8 ± 17.1	b	
K_4%_48	132.2 ± 27.7	ab	9.0 ± 0.0	c	125.1 ± 10.1	a	K_4%_48	185.8 ± 26.6	a	7.0 ± 0.0	c	131.6 ± 21.2	ab	
K_4%_96	123.7 ± 23.5	b	10.0 ± 0.0	bc	121 ± 17.5	a	K_4%_96	81.18 ± 15.7	c	9.0 ± 0.0	b	59.6 ± 27.5	cd	
K_6%_24	87.11 ± 14.6	c	10.0 ± 0.0	bc	83.77 ± 16.2	bc	K_6%_24	122.9 ± 27.5	bc	8.0 ± 0.0	bc	104.3 ± 24.9	b	
K_6%_48	123.3 ± 18.1	b	9.1 ± 0.4	c	124.1 ± 18.2	a	K_6%_48	141.3 ± 24.7	b	8.0 ± 0.0	bc	104.1 ± 21.2	b	
K_6%_96	179.0 ± 32.6	a	7.0 ± 0.0	c	144.4 ± 16.9	a	K_6%_96	216.5 ± 35.1	a	7.0 ± 0.0	c	150.2 ± 32.9	a	
	C	100.0 ± 0.0	c	16 ± 0	a	36.3 ± 15	c		C	100.0 ± 0.0	c	12.0 ± 0	a	37.0 ± 12.8	d	
Note:

* Different letters indicate significant differences according to Tukey’s HSD test.

Cultivar-specific differences were observed in DGI. This index reached the highest values in PY/PR; specifically, in PR and PY, DGI under KNO3 6% for 96 h, was about twice the control 179.0 ± 32.6% and 216.5 ± 35.1%, respectively. In Sassaniello instead the highest values for DGI were >160%, and were observed in SR under hydropriming 24 h, KNO3 2% for 24–48 h and KNO3 6% for 96 h, while in SY, under hydropriming 24 h and KNO3 6% for 96 h. Also, in SY, PR, and PY a reduction in DGI of about 20% compared to control was sometimes observed due to a reduced root length (see Table 1).

Antioxidant efficacy

For antioxidant evaluation, only priming treatments providing the best germination and growth performance at each individual cultivar were considered and compared to control. The selected treatments were common to all four cultivars; they were hydropriming for 24 h, KNO3 4% for 48 h and KNO3 6% for 96 h (Fig. 2). In PR control antioxidant content in leaf tissue was lower (about halved) than PY, and no significant difference was observed between the antioxidant concentration in leaf from primed seeds in comparison with control. In PY instead, KNO3 6% for 96 h significantly increased antioxidants in leaf tissue. In SR both hydropriming and even more KNO3 6% for 96 h significantly increased antioxidant concentration compared to control. In SY all priming treatments significantly increased the concentration of antioxidant in the leaves, with KNO3 6% > KNO3 4% > H.

Figure 2 Average total antioxidant content.

Bar charts of average total antioxidant content measured in the leaf tissue at 40 days culture in papaccella ((A), PR and PY) and sassaniello ((B), SR and SY) for control (C), hydropriming (H-24 h) and halopriming (KNO3 4% and 6% at 48 and 96 h, respectively) treatments. The error bars are standard deviations. Different letters indicate significant differences according to Tukey’s HSD test (p < 0.05).

Protein content

In SR seeds, all priming treatments significantly stimulated the production of protein in comparison with control, with hydropriming which approximately tripled it (Fig. 3). Similarly, in SY all priming treatments increased protein content, but in this case KNO3 6% for 96 h showed the best performance, with a protein content almost quadrupled compared to control. In PY and PR all treatments significantly increased protein content in seeds, with a similar trend in both cultivars: i.e., KNO3 6% > Hydropriming > KNO3 4%

Figure 3 Average protein content.

Bar charts of average protein content measured papaccella ((A), PR and PY) and sassaniello ((B), SR and SY) for control (C), hydroprimed (H-24 h) and haloprimed (KNO3 4% and 6% at 48 and 96 h, respectively) seeds. The error bars are standard deviations. Different letters indicate significant differences according to Tukey’s HSD test (p < 0.05).

Discussion

The present work evidenced that all priming treatments investigated improved seed germination and plant vigor in the four cultivars examined. These results are especially relevant for some reasons: firstly, pepper is one of the most important vegetable crops worldwide, because of its adaptation to different agroclimatic regions and the wide variety of shapes, sizes, colors, and tastes of the fruits (Ozbay, 2018; Qin et al., 2014). Moreover, pepper is suitable for traditional open field cultivation, but is also an important product of greenhouse cultures (Qiu et al., 2009), which have become a relevant part of plant food production, guaranteeing an annual supply of vegetables, regardless the season. Finally, Capsicum species are known for their poor seed germination and longevity, shorter than other Solanaceae (Bissoli et al., 2022). This can explain the research interest in developing systems to improve seed quality and germination performance, which has been going on for over 40 years (e.g., (Smith & Cobb, 1991; Sundstrom & Edwards, 1989; Yaklich & Orzolek, 1977)). The worse germination performance of pepper compared to other Solanaceae (e.g., tomato and eggplant) is at least partly due to the seed coat which is thick, determining a slow and irregular germination (Qiu et al., 2009). To cope with this issue, we also tested acid priming, hypothesizing that acid could weaken the seed coat, as it happens during digestion process. In this respect, contrasting results are reported in the literature; some authors hypothesized that acid treatments, miming the digestion process occurring in the birds, could contribute to breaking dormancy, and decreasing MGT. Others found a detrimental effect of digestion on pepper seed germination. These latter authors conclude that the positive effect related to birds could be due to dispersal of the seeds in favorable environments rather than to digestion itself (Quintero et al., 2018 and references therein). Our results, indicate that the acidic primer completely inhibited germination, likely because the slight opening of the hilum allows the hydrochloric acid solution to cross the integuments without deteriorating them, reaching the embryo, and damaging it (Fig. S1).

All the other priming methods here investigated significantly decreased the MGT, but the best results were overall obtained by KNO3 6% for 96 h. It is widely assessed that the activation of pre-germinative metabolism in seeds requests a progressive water absorption (Paparella et al., 2015; Rajjou et al., 2012); it is likely that this concentration of salt, together with the relatively long treatment time, could guaranty an entry of water slow enough to obtain uniform hydration.

The best response of the four cultivars to priming with KNO3 6% for 96 h can also find an easy explanation in its role as a source of nutrients, namely nitrate and potassium, which improve the activity of α-amylase, as demonstrated in various crop species (Sher et al., 2019).

Apart from the study by Smith & Cobb (1991) who found a final germination percentage of pepper seeds primed with KNO3 lower than control (88% vs. 95%), our data are overall in agreement with previous published works on pepper seed priming. In fact, Venkatasubramanian & Umarani (2010) found that 5% KNO3 for 24–48 h had a positive effect on percentage and speed of germination in chilli pepper, as well as in plant growth evaluated as seedling length. Kumar, Chaurasia & Marmat Sandip (2021) testing several inorganic and organic priming methods in chili pepper, found that 2% KNO3 was one of the best pre-sowing treatments with a positive effect on seed germination and plant growth. Tu et al. (2022) highlighted that 3% KNO3 for 48 h increased germination percentage, fresh weight, and SVI in three pepper cultivars, with a better performance in comparison with 20% PEG and combined treatment (PEG + KNO3).

Derived germination indexes SVI and DGI, put the germination percentage in relation with other parameters measured on seedlings. Therefore, following the early development of plantlets we can understand if a higher germination percentage or a lower MGT are associated to an improvement in plant development. In our case, the significant increase of both DGI and SVI in plants from primed seeds in all the four cultivars indicates that priming, and especially 6% KNO3 for 96 h, positively affected both root and shoot growth. Under the best treatments DGI, which is expressed as percentage compared to control (set to 100%) increased about 70% in sassaniello cultivars and 70–100% in red and yellow papaccella. As for T50 values, they were lower (i.e., indicating a fast germination) in the seeds primed with 4% KNO3 for 48 h and 6% KNO3 for 96 h in all cultivars. Accordingly, quite comparable MGTs, were calculated in both seeds primed with 4% KNO3 for 48 h and 6% KNO3 for 96 h (Fig. 1), indicating that both treatments had similar effect on germination. Nonetheless, in seeds primed with KNO3 6% for 96 h this faster germination reflected on DGI, SVI, and growth parameters; this suggests that other factors could play a role in germination performance and plant growth. In fact, it is well assessed that priming increases protein concentrations in seeds, especially α-amylase, involved in starch mobilization and use, and ROS-scavengers, as catalase, to protect plasma membrane from oxidative damage (e.g., (Arun et al., 2017; Waqas et al., 2019)). In line with this evidence, our results highlight that the treatment KNO3 6% for 96 h maximized the production of proteins in seeds (in PR, PY, and SY) up to 8 times greater than the control (in PR), which confirms a rapid activation of the pre-germinative metabolism. Similarly, it is reported that seed priming increases total antioxidant concentration in seedlings of numerous species (e.g., (ElSayed et al., 2022; Kasote et al., 2019; Shah et al., 2021)). In all cultivars, except for PR, total antioxidants significantly increased at the maximum concentration under the same priming treatment (i.e., KNO3 6% for 96 h); in parallel, the same halopriming significantly increased also protein content in the seeds. Therefore, the most performant priming treatment, 6% KNO3 for 96 h, speeded up seed germination, probably by a faster activation of pre-germinative metabolism, maximizing protein production in seeds and increasing antioxidant production in leaf tissue, with positive effect on plant growth.

Seed germination and first stages of plant development are critical steps in plant life, in which the embryo must be able to interrupt dormancy, restore metabolism, and promote plant establishment. This process requires multiple morpho-physiological events in which modification of gene expression, and subsequent activation of specific biochemical pathways, result finally in proper morphological responses, such as primary root emergence, sprout development and stable seedling establishment (Paparella et al., 2015; Sharma et al., 2022). During this phase the seedling is more vulnerable and therefore more exposed to biotic and abiotic stress (Gull, Lone & Wani, 2019; Sorrentino et al., 2024; Zulfiqar, 2021). Speeding up this phase, means reducing these risks and moving quickly towards a phase with greater survival potential, i.e., the adult plant, in which life machinery is fully operational.

Conclusion

This work, focused on seed priming techniques in four ancient and neglected cultivars of Capsicum annuum, highlights that both hydro- and halopriming improved germination performance and plant growth, confirming that seed priming is a valuable, ecofriendly method to reawaken pre-germinative metabolism, increasing seed quality and crop yield. The treatment with KNO3 6% for 96 h in addition to improving germination performance, stimulated plant growth with significant increase of biomass production. Further, it significantly enhanced protein content in the seeds of all four cultivars and leaf concentration of total antioxidants in three cultivars. Finally, the results described will be a good starting point for testing the effectiveness of these priming treatments to improve germination in aged seeds, as well as the ability of primed seeds to adapt to poor or exploited soils.

Supplemental Information

Supplemental Information 1 Seed vitality test.

Pepper seeds (cut in half) stained by tetrazolium solution: a,b) Control seeds, a) viable and vigorous, b) viable non-vigorous; c,d) Priming with KNO3 6% 96h , c) viable and vigorous, d) viable non-vigorous; e,f)Acid priming with HCl at two different times, e) 5’ non-viable; f)10’ non-viable.

Supplemental Information 2 Summary of the general linear models (GLMs) testing for main and interactive effects of cultivar (CV), Treatments and Time for the viables.

MGT, morphological data, SVI, DGI and T50. P-values < 0.05 are reported in italic.

Supplemental Information 3 Data.

Additional Information and Declarations

Competing Interests

Author Contributions

Data Availability

Fiore Capozzi and Valeria Spagnuolo are Academic Editors for PeerJ. Riccardo Riccardi and Patrizia Spigno are employed by Cooperativa ARCA 2010.

Angelo Granata conceived and designed the experiments, performed the experiments, prepared figures and/or tables, authored or reviewed drafts of the article, and approved the final draft.

Fiore Capozzi performed the experiments, analyzed the data, prepared figures and/or tables, authored or reviewed drafts of the article, and approved the final draft.

Anna Gaglione performed the experiments, authored or reviewed drafts of the article, and approved the final draft.

Riccardo Riccardi performed the experiments, prepared figures and/or tables, and approved the final draft.

Patrizia Spigno performed the experiments, prepared figures and/or tables, and approved the final draft.

Simonetta Giordano conceived and designed the experiments, prepared figures and/or tables, authored or reviewed drafts of the article, and approved the final draft.

Maria Cristina Sorrentino performed the experiments, authored or reviewed drafts of the article, and approved the final draft.

Valeria Spagnuolo conceived and designed the experiments, analyzed the data, prepared figures and/or tables, authored or reviewed drafts of the article, and approved the final draft.

The following information was supplied regarding data availability:

The raw data are available in the Supplemental File.

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
