# Peer review of "Seed priming enhances seed germination and plant growth in four neglected cultivars of Capsicum annuum L"

_PeerJ, doi:10.7717/peerj.18293_

## Round 0.1 · original submission · Major Revisions

Dear Authors,
Please improve your manuscript by addressing the issues of the Reviewers.

Reviewer 1 ·

Basic reporting

In this study the authors investigated the effects of seed priming on four endangered varieties of Capsicum annuum, revealing that both hydropriming and halopriming with potassium nitrate (KNO3) significantly enhanced germination rates and subsequent plant growth, claiming it a promising approach for improving the germination rate.

The article has a few problems with the writeup. Major one identified as run-on sentences i.e sentences that are very long containing more than one idea in one sentence and losing the reader's flow and attention.

Secondly, the article seems to be translated from another language and does not comply with the correct sentence structure. e.g

Introduction first sentence is poorly constructed
Intro 2nd Para 3rd Sentence quotes "Product competitiveness" which seems out of place or not fit? and many other instances.
Therefore article does not meet the language standards.

Formulas should be inserted as an 'equation' not as 'text'...!

Experimental design

It is unclear why the cultivars used in the study are endangered or threatened and what is their importance.
Seed was acquired in 2010. It is not mentioned in the article when the experiment reported was conducted. Nonetheless, the materials and methods used reference of years 2021 and 2022. This suggests that the experiment was conducted after 2022. (The simple vigor index (SVI) Tu et al. 2022; Wang et al. 2021 etc)
Having said that the experiment was conducted at least 12 years after the seed was acquired. This itself is a big question mark especially reporting studies on germination percentage. The authors themselves reported, "Finally, Capsicum species are known for their poor seed germination and longevity, shorter than other Solanaceae (Bissoli et al. 2022)".
Old seeds can have poor germination due to several reasons:

Reduced viability: Seeds have a limited lifespan, and their ability to germinate decreases over time. Moisture content: Low moisture content can cause seeds to become dormant or non-viable.
Storage conditions: Improper storage, such as exposure to heat, light, or moisture, can damage seeds.
Pathogens and pests: Old seeds may be more susceptible to pathogens and pests, which can reduce germination. Hormonal changes: Changes in hormone levels within the seed can affect germination.
Enzyme degradation: Enzymes necessary for germination may break down over time.

Based on these confounding effects the results inferred will be very unlikely.

Validity of the findings

The authors wrote different concentrations of the KNO3 (2%, 3%, and 4%) working best to improve germination by other scientists in the review of literature. However, their conclusion exclusively considered 6% application as the best treatment.
The authors also concluded their results for other cultivars. However, the cultivars can have diverse backgrounds and such claims are very optimistic. Rather the results should be inferred about the treatment. Their claim for worldwide application by quoting:
“Remarkably, the same treatment was the best in all four cultivars here examined, suggesting its possible efficacy also in other cultivars of Capsicum annuum, an important result considering that this crop species is known and cultivated worldwide.”
Employing the results on very shaky grounds and cannot be inferred for the entire globe.

Reviewer 2 ·

Basic reporting

I have gone through the MS entitled “Seed priming enhances seed germination and plant 1 growth in four threatened cultivars of Capsicumannuum L.” The MS is well written and presented, however there is great room for improvements as below;
Although there are less errors in English, English language expression needs to be improved e.g., in lines 39-40, and other places in the MS.
“Therefore, it is mandatory to develop alternative practices allowing a high yield of agricultural products, but in the name of sustainability and the responsible use of resources.”
Reference list need to be revised as per PEERJ. Many scientific words are not italic. See Journal guidelines.
Revisit tables for lettering/statistics or recompose to clarify wrong presentation.
Revisit graphs for better presentation like lettering need to be close to bars. Bar width needs to be increased. Resolution is not good.
Raw data provided. Spell check all the raw data e.g., Root lenght or Root length?
Line 46 and many other places, referencing is not as per PEERJ
(Jisha et al. 2013; Paparella et al. 2015; Rajjou et al. 2012). and so many others
Materials and Methods need to be revised as there are some irrelevant information like the text under heading ‘Plant material’ that information can be part of introduction
There are many formatting errors like expression of temperature
Line 116= 5’, 10’ and 15’, ?
In Morphological observations, no reference was given, Authors are encouraged to see the literature
Line 203; for 10’.?
3.4. Antioxidant power should be Antioxidant efficacy
The authors need to clarify the evidence behind “four endangered cultivars of Capsicum annuum” reference ?
In Conclusion section, methodology needs to be avoided
Line 387-388, This treatment in fact, in addition to improve germination performance, …
should be as ‘This treatment in fact, in addition to improving germination performance,…

Experimental design

The article falls within the aims and scope of PEERJ. Research question not well defined. Methods described with sufficient details.

Validity of the findings

Novelty not well clear. Meaningful replication done as seen in the raw data.
All underlying data have been provided; they are robust, statistically sound, & controlled.
Conclusions are well stated, linked to original research question & limited to supporting results. The authors need to clarify the evidence behind “four endangered cultivars of Capsicum annuum”.

---

## Round 0.2 · Minor Revisions

Please improve graph quality. Please cite some manuscripts about the status of these endangered species.

Reviewer 1 ·

Basic reporting

The authors improved the manuscript significantly and is now in acceptable form.

Experimental design

There was a dubious statement on Seed acquisition which is clarified now, and the study makes sense as the Seed was acquired just before experimentation.

Validity of the findings

Looks alright.

Reviewer 2 ·

Basic reporting

My all previous comments/ suggestions have not been addressed fully. For instance graph quality have not been improved.

Experimental design

-

Validity of the findings

Evidence on the cliam about 'endangered species' not clear yet. Cit some papers about the status of these endangered species. Just showing the definition of endangered species will not be enough.

---

## Round 0.3 · accepted · Accept

Authors have addressed all of the reviewers' comments. I am happy with the current version. This manuscript is ready for publication.

Reviewer 1 ·

Basic reporting

My Comments were already addressed.

Experimental design

Just fine.

Validity of the findings

Looks Ok.

Reviewer 2 ·

Basic reporting

The authors addressed the issues

Experimental design

-

Validity of the findings

-